# Knowledge, attitudes, practices of/towards COVID 19 preventive measures and symptoms: A cross-sectional study during the exponential rise of the outbreak in Cameroon

Adela Ngwewondo[1]*, Lucia Nkengazong[1], Lum Abienwi Ambe[1], Jean Thierry Ebogo[1], Fabrice Medou Mba[1], Hamadama Oumarou Goni[1], Nyemb Nyunaï[1], Marie Chantal Ngonde[1,2], Jean-Louis Essame Oyono[1]

**1** Institute of Medical Research and Medicinal Plants Studies (IMPM), Centre of Medical Research, B.P.6163 Yaoundé, Cameroon, **2** Centre Hospitalo-Universitaire, Yaoundé, Cameroon

* adelafopezi@yahoo.fr

## Abstract

Severe Acute Respiratory Syndrome Coronavirus 2 (COVID 19) has plagued the world with about 7,8 million confirmed cases and over 430,000 deaths as of June 13th, 2020. The knowledge, attitude, and practices (KAP) people hold towards this new disease could play a major role in the way they accept measures put in place to curb its spread and their willingness to seek and adhere to care. We sought to understand if: a) demographic variables of Cameroonian residents could influence KAP and symptomatology, and b) KAP could influence the risk of having COVID19.A cross-sectional KAP/symptomatology online survey was conducted between April 20 to May 20. All analyses were performed using SPSS version 23. Of all respondents (1006), 53.1% were female, 26.6% were students, 26.9% interacted face to face and 62.8% were residents in Yaoundé with a median age of 33. The overall high score was 84.19% for knowledge, 69% for attitude, and 60.8% for practice towards COVID 19. Age > 20 years was associated with a high knowledge of COVID 19. Women had lower practice scores compared to men (OR = 0.72; 95%CI 0.56–0.92). 41 respondents had ≥3 symptoms and only 9 (22.95%) of them had called *1510* (emergency number). There was no significant difference between KAP and symptomatology. The presence of ≥ 3 symptoms in 4% of respondents (with 56% of them having co-morbidities) supports the current trend in the number of confirmed cases (8681) in Cameroon. The continuous increase in the number of cases and the overall good KAP warrants further investigation to assess the effectiveness of the measures put in place to curb the spread of the disease. Sensitization is paramount to preclude negative health-seeking behaviors and encourage positive preventive and therapeutic practices, for fear of an increase in mortality.

## Author summary

SARS-COV-2 is transmitted from person-to-person through inhalation of aerosols from an infected individual. Old age and patients with pre-existing illnesses (like hypertension,

---

**Data Availability Statement:** All relevant data are within the manuscript and its Supporting Information files.

**Funding:** The authors received no specific funding for this work.

**Competing interests:** The authors have declared that no competing interests exist.

cardiac disease, cancer, or diabetes) have been identified as potential risk factors for severe disease and mortality. More information about its distribution, transmission, pathophysiology, treatment, and prevention are needed. World Health Organization (WHO) recommends the prevention of transmission by using face masks, washing hands, and social distancing. We investigated the way people accept measures to curb the spread of disease and their willingness to seek and adhere to care when presenting symptoms. The knowledge of COVID 19 mode of transmission was satisfactory among the Cameroonian population. Most respondents had high practice scores towards preventive measures and positive health-seeking behaviors, although a few stigmatized the hospital milieu and resorted to auto-medications/ traditional concoctions. However, the continuous increase in the number of cases and the overall high KAP scores warrants further investigation to assess the effectiveness of the measures put in place. Also, the presence of COVID 19 symptoms in the population adds more evidence to active disease transmission in the community. This calls for widespread testing in the community because <22% of people with COVID 19 symptoms seek help.

## Introduction

According to the World Health Organization (WHO), viral diseases continue to emerge and represent a serious issue to public health. In the last twenty years, several viral epidemics such as the severe acute respiratory syndrome coronavirus (SARS-CoV) from 2002 to 2003, and H1N1 influenza in 2009, have been recorded. Most recently, the Middle East respiratory syndrome coronavirus (MERS-CoV) was first identified in Saudi Arabia in 2012 and now the new Coronavirus disease 2019 (COVID-19) has plagued the world. COVID-19 is an emerging respiratory disease caused by the highly contagious novel coronavirus (SARS-CoV 2) and was first detected in December 2019 in Wuhan, China [1–3]. This new virus has quickly spread globally afflicting 215 countries. As of June 13th, 2020, over 7.8 million cases and 430,000 deaths have been reported globally [4].

In Cameroon, the first case confirmed on the 6th of March 2020 was a French national who arrived Yaoundé. To control and avoid the rapid spread of the ongoing COVID-19 outbreak in the country, several measures were adopted by the government. These measures include the limitation of the number of passengers of public transportation; the closing of all schools; quarantine and care for infected people or suspected cases; closure of borders and suspension of flights; suspension of issuance of entry visas to Cameroon; gatherings of more than 50 people prohibited; bars, restaurants, and public places closed from 6 pm; consumer flow regulations set up in markets and supermarkets; urban and inter-urban travel only undertaken in cases of extreme necessity; overloading in public transport vehicles prohibited; implementation of virtual meetings; avoiding close contact such as shaking hands or hugging and covering the mouth when sneezing.

The number of those infected is still on the rise and after the government uplifted some of the COVID-19 restrictive measures on bars, taxis, restaurants. Several other cases were confirmed later amounting to 8681 cases, 4836 recovered and 212 deaths [4]. The infection rate and the resources needed to battle this disease can be expected to increase exponentially.

Currently, there is no approved treatment for COVID-19 and no clinical trial data supporting any prophylactic treatment. In the absence of this approved treatment, available treatments are directed at relieving symptoms and the panic of no approved treatment can lead to embracing other non-standard options. Agents previously used to treat SARS and MERS are

potential candidates to treat COVID-19 [5] as well as various agents with apparent in vitro activity against severe acute respiratory syndrome coronavirus (SARS-CoV) and Middle East respiratory syndrome coronavirus (MERS-CoV). These include chloroquine and hydroxy-chloroquine, also used in the prevention and treatment of malaria and/or chronic inflammatory diseases. Remdesivir, an antiviral drug is being touted as a possible coronavirus treatment [6].

Responses to epidemics in Africa have been challenged by limited infrastructure and fragile healthcare systems. This includes the lack of adequate surveillance to assess the scope of the outbreak, and inadequate systems for the prevention, diagnosis, and management of a disease. Cases of COVID 19 as with other diseases are broadly classified as suspected, probable, and confirmed [7]. Assessing the symptoms of COVID 19 (suspected cases) is a preliminary step in the diagnosis and management of this disease.

Person to person transmission (community spread) is currently ongoing in the country, making it necessary to control the disease to avoid its rapid spread throughout the country. To guarantee successful disease control, people's adherence to preventive and control measures are essential. This adherence is highly dependent on the population's knowledge, attitudes, and practices (KAP) towards COVID-19 following the KAP theory. A previous study indicates that the knowledge level and attitudes towards infectious diseases are associated with the level of panic among the population, which can further complicate attempts to prevent the spread of the disease, thus prompting alternative treatment sources [8].

Due to shortages in diagnostic kits in our setting during our study, a questionnaire to gauge the symptomatic trend was issued to Cameroonians to strategically define an approach to address the outbreak. In this study, according to guidelines for clinical and community management of COVID-19 by the Cameroon National Health Development Plan, our objective was to evaluate the factors influencing the knowledge, attitudes, and practices of Cameroonian respondents on COVID 19 and also evaluate the associations between the demographics, KAP and symptoms of COVID 19.

## Methodology

### Ethical consideration

The Ethical Committee of the Institute of Medical Research and Medicinal Plants Studies, Yaoundé, approved the study protocol and procedures before the formal survey.

### Study design

This cross-sectional survey was conducted from April 20 to May 20, two weeks after the partial confinement was implemented in the country. Because it was not feasible to do a community-based national sampling survey during this period, data was collected online. Relying on the authors' networks with local people living in all the 10 regions of Cameroon (S1 Table), a one-page recruitment sheet was posted to groups and individuals via "WhatsApp", email, websites accounts. This page (S1 Appendix) contained a brief introduction to the background, objective, procedures, voluntary nature of participation, declarations of anonymity and confidentiality, and notes for filling in the questionnaire, as well as the link of the online copy questionnaire. The associations between the demographics and KAP, symptoms and KAP, comorbidities, symptoms of COVID 19 with age, and gender were assessed.

The independent variables were symptomatology, demographic characteristics (gender, age, profession, working environment, city of residence).

The dependent variables were knowledge, attitudes, practice, and symptoms of/towards COVID 19. Questions on knowledge were about the mode of transmission, attitude towards

health-seeking behaviors and practices like avoiding crowded areas, wearing masks, washing hands and using sanitizers, taking vitamin C, citrus fruits, traditional concoctions, and drugs (chloroquine, ibuprofen, paracetamol). Symptomatology was assessed by someone presenting with fever, headache, dry cough/catarrh, throat irritation, diarrhea, difficulty breathing, and muscle pain. The symptomatology section was accompanied by associated comorbidities and diseases with similar symptoms.

## Study population and eligibility criteria

Cameroonian residents, aged 18 years or more, employed or unemployed who understood the content of the recruitment page- and who agreed to participate in the study completed the questionnaire.

## Measures

The questionnaire consisted of four parts: demographics, knowledge, attitudes, practices, and symptomatology. A COVID-19 KAP and symptomatology questionnaire was developed. The questionnaire had 32 questions: 7 items on knowledge, 4 items on attitudes, 9 items on practices, 5 items on symptomatology (for suspected cases), 5 on demographics, and 2 others (source of information). Questions were answered on a Yes/No basis with an additional "I don't know" option. Some open-ended questions were asked.

## Statistical analysis

A score of 1 was attributed to a correct answer and 0 to a wrong answer for knowledge, attitude, and practice. The Knowledge range was 3–7, 1–4 for attitude, and 2–9 for practice. The overall scores of each individual were used to obtain mean scores for KAP. Blooms' cut-off was used. Frequencies of correct knowledge answers and various attitudes and practices were described. Multiple logistic regression analysis using all of the demographic variables as independent variables and knowledge/ practice score as the outcome variable was conducted to identify factors associated with knowledge and practice. Similarly, binary logistic regression analyses were used to identify factors associated with practices. Factors were selected with a backward stepwise method. Unstandardized regression coefficients ($\beta$) and odds ratios (ORs) at 95% confidence intervals (CIs) were used to quantify the associations between variables and KAP. Associations between demographic variables of gender and age were compared to comorbidities and symptoms ($\geq$3). Also, associations between demographics and KAP were studied. Data analyses were conducted with SPSS version 23.0. The statistical significance level was set at $p < 0.05$ (two-sided).

## Results

### Socio-demographic characteristics

A total of 1006 participants completed the survey. The average age of those who participated was 33±11.2. The demographic characteristics of all the participants are presented in Table 1. Of all respondents, 53.1% were female and 26.6% were students in the university or high school. About the working environment, 26.9% of the participants interacted face to face with others (i.e customer services cashiers, hairdressers, traders, etc), 25.5% spent most of their time in the office, and 20.4% at home. Overall, 62.8% were residents of the capital city Yaoundé.

**Table 1. Demographic characteristics of participants.**

| Demographics/characteristics | Number of respondents | Percentage (%) |
|---|---|---|
| **Gender** | | |
| • **Male** | 472 | 46.9 |
| • **Female** | 534 | 53.1 |
| Age | | |
| • **<20** | 32 | 3.2 |
| • **[20–30]** | 361 | 35.9 |
| • **[30–40]** | 367 | 36.5 |
| • **[40–50]** | 139 | 13.8 |
| • **≥50** | 107 | 10.6 |
| **Profession** | | |
| • **Health care worker** | 99 | 9.8 |
| • Private sector worker | 182 | 18.2 |
| • **Public service personel** | 141 | 14 |
| • Retired | 17 | 1.7 |
| • **Student** | 268 | 26.6 |
| • **Teacher/Lecturer** | 135 | 13.4 |
| • **Others (housewives, farmers, unemployed, taxi drivers, builders)** | 164 | 16.3 |
| **Working environment** | | |
| • **At home** | 205 | 20.4 |
| • **Face to face interaction with customers** | 271 | 26.9 |
| • **Hospital** | 102 | 10.1 |
| • Office | 257 | 25.5 |
| • **Outdoor environment** | 171 | 17.1 |
| **City of residence** | | |
| • **Yaoundé** | 632 | 62.8 |
| • **Douala** | 146 | 14.5 |
| • **Buea** | 89 | 8.8 |
| • **Others (cities from the 10 regions of Cameroon)** | 139 | 13.9 |

## Source/ period of information on COVID 19

For the source/period of information, 41.9% of the respondents knew when the disease began (December 2019) and 14.4% only knew in March 2020 when the first case was reported in Cameroon. Greater than half of the respondents (54.5%) got the information on COVID 19 for the first time via the television during the first and last 15 days of the study, the respondents got the information primarily through television followed by Whatsapp and websites (Table 2).

## Knowledge related to COVID 19

For the mode of transmission, 94.3% knew that the disease could be transmitted by droplets when an infected person coughs, sneezes or speaks, 75.6% said through kissing infected person, 74.7% through a handshake, 88.3% through touching a contaminated surface and then touching your eyes, nose or mouth (Table 3). The results show that 84.19% (n = 847) respondents had high knowledge score of 4–7 on the transmission of the disease (Table 4).

## Attitudes towards COVID 19 pandemic

Attitude towards COVID 19 health-seeking behaviors was assessed. Of all respondents, 73.1% think they can be contaminated by health care workers, 28.6% refuse to go to the hospital even

**Table 2. Source/ period of information on COVID 19.**

| Month/Source of information | Mouth to mouth | Newspaper | Television | Websites | Whatsapp | Total |
|---|---|---|---|---|---|---|
| DEC-19 | 25(6.0%) | 27(6.4%) | 246(58.7%) | 77(18.4%) | 44(10.5%) | 419 |
| JAN-20 | 14(5.1%) | 13(4.8%) | 141(51.6%) | 46(16.8%) | 59(21.6%) | 273 |
| FEB-20 | 19(11.2%) | 6(3.6%) | 84(49.7%) | 26(15.4%) | 34(20.1%) | 169 |
| MAR-20 | 28(19.3%) | 7(4.8%) | 77(53.1%) | 13(9.0%) | 20(13.8%) | 145 |
| Total | 86(8.5%) | 53(5.3%) | 548(54.5%) | 162(16.1%) | 157(15.6%) | 1006 |
| Source of information of respondents within the first and last 15 days of study period | | | | | | |
| First 15 days of study | 50(7.2%) | 34(4.9%) | 370(53.3%) | 123(17.7%) | 117(16.9%) | 694 |
| Last 15 days of study | 36(11.5%) | 19(6.1%) | 178(57.1%) | 39(12.5%) | 40(12.8% | 312 |
| Total | 86(8.5%) | 53(5.3%) | 548(54.5%) | 162(16.1%) | 157(15.6%) | 1006 |

if they are suffering from another disease other than COVID 19. Out of the 28.6% of those who do not want to go to the hospital, 21.3% are afraid of being contaminated in the hospital with nosocomial infections like COVID19, 3.5% think the health personnel can misdiagnose their illness given that many other diseases have similar symptoms (Table 3).

Regarding people's willingness to do a COVID 19 test, 72.0% were willing to do a voluntary test among which 47% of them preferred the house over the hospital for their medical care if tested positive. People's preference for house medical care is because they are afraid of being contaminated in the hospital (38.6%), their families can take good care of them and they feel comfortable at home (58.1%) (Table 3). Overall, 69% of respondents had a high attitude score of 2–4 for health-seeking behavior (Table 4).

## Practices towards COVID 19 pandemic

All the respondents use masks (100%), 94.5% wash hands and use sanitizers, 83.8% practice social distancing, or don't go to crowded places, while 20% are confined at home. Looking at what people take as preventive measures, 74.6% eat citrus fruits and take vitamin C tablets. We also observed that 35.9% resort to traditional concoctions, auto medications like chloroquine (4.4%), (5.6%) paracetamol, and Ibuprofen (Table 3).

Also, 60.8% of respondents were taking precautions (good practice) like avoiding crowded areas, wearing masks, washing hands, and using sanitizers, taking vitamin C and citrus fruits. Statistical significant differences in knowledge of disease transmission were observed with gender, working environment, and city of residence (Table 5).

Age > 20 years was associated to high knowledge and attitude scores on COVID 19 (Table 6). Women had lower practice scores compared to men (OR = 0.72; 95% CI 0.56–0.92) and Douala respondents had had high practice scores (OR = 1.16; 95%CI 1.13–2.45) when compared to those from Yaoundé (Table 6).

## Symptomatology and associated co-morbidities of COVID 19

Out of all the respondents, 71.7% reported no symptoms of the disease, while 5.0% reported fever, 8.3% dry cough/catarrh, 6.5% throat irritation, 13.1% headache, 0.9% diarrhea, 2.5% difficulty breathing, 6.0% muscle pain and 2.2% don't smell odor or taste. Of all the respondents, 4.7% suffered from hypertension, 6.9% from a common cold, and 24.6% from malaria (Table 7). Significant statistical differences were observed with respondents having hypertension > 30 years when compared to those <30 years (Table 7). Forty-one respondents (4.1%) had more than 3 symptoms with only 9 (21.95%) who called *1510* while 32 (78.05%) did not call *1510*.

**Table 3.  Knowledge, attitudes, and practices of COVID 19 among the 1006 respondents in Cameroon, n (%).**

| Questions/ options | Yes (%) | No (%) |
|---|---|---|
| **Knowledge of transmission (How is COVID 19 transmitted?)** | | |
| K1. Droplets when an infected person coughs, sneezes or speaks | 941 (94.3) | 5.7 (57) |
| K2. Kissing an infected person | 761 (75.6) | 245 (24.4) |
| K3. Handshake | 751 (74.7) | 255 (25.3) |
| K4. Touching a contaminated surface and then touching your eyes, nose or mouth | 888 (88.3) | 118 (11.7) |
| K5. Blood transfusion[F] | 77 (7.7) | 929 (92.3) |
| K6. Sexual intercourse[F] | 74(7.4) | 932 (92.6) |
| K7. Contaminated foodstuffs | 377 (37.5) | 629 (62.5) |
| **Attitude towards COVID 19 health-seeking behavior** | Yes (%) | No (%) |
| A1. If you are living with someone working in a hospital milieu do you think they can contaminate you? | 735 (73.1) | 271(26.9) |
| A2. Are you willing to do a voluntary test for COVID 19? | 724 (72) | 282 (28) |
| A3. If you have another disease other than COVID 19, will you go to the hospital? | 718 (71.4) | 288 (28.6) |
| If No, why won't you go to the hospital? | | |
| • Afraid of contamination | 214 (21.3) | / |
| • Misdiagnosis | 35 (3.5) | / |
| • Self-treatment | 13 (1.3) | / |
| • To avoid stigmatization | 9 (0.9) | |
| A4. Do you prefer to be confined in the house or hospital for your medical care when you are tested positive for COVID 19? | | |
| • House | 473 (47) | / |
| • Hospital | 378 (37.6) | / |
| •I don't know | 155 (15.4) | / |
| Please explain the choice of staying at home | | |
| • Afraid of contamination | 166 (38.6) | / |
| • Better home care with family | 250 (58.1) | / |
| • Less costly | 4 (0.9) | / |
| • Stigmatization | 10 (2.3) | / |
| **Practice on preventive measures** | | |
| P1. Social distancing | 843 (83.8) | 163 (16.2) |
| P2. Washing hands and using sanitizers | 951 (94.5) | 55 (5.5) |
| P3.Total confinement | 201 (20) | 805 (80) |
| P4. Use of mask | 1006 (100) | 0 (0) |
| P5. Use of Traditional concoctions[F] | 361 (35.9) | 645 (64.1) |
| P6. Taking chloroquine[F] | 44 (4.4) | 962 (95.6) |
| P7. Eating citrus fruits such as lemon and taking Vitamin C tablets | 746 (74.6) | 260 (25.4) |
| P8. Taking paracetamol[F] | 46 (4.6) | 960 (95.4) |

(*Continued*)

**Table 3.** (Continued)

| Questions/ options | Yes (%) | No (%) |
|---|---|---|
| P9. Taking Ibuprofen[F] | 10 (1.0) | 996 (99) |

[F]False answers

The association between the KAP and the symptomatology was assessed to see the influence of KAP on the risk of having the disease. Interestingly, there were no significant differences between those having symptoms and those without symptoms. Also, no significant difference was observed with those having <3 symptoms to those having >3 symptoms (Table 8).

Overall 23 respondents having ≥4 symptoms had underlying comorbidities and 10 of them had other diseases with symptoms similar to COVID 19 (Fig 1). Also, 168 (18.1%) out of 262 of those with symptoms did not have them recurrently.

**Table 4. Associations between background characters and Knowledge, attitudes regarding COVID 19.**

| Variable | Knowledge Score (4–7) | Knowledge Score (0–3) | (P-value) | Attitude Score (2–4) | Attitude Score (0–2) | (P-value) |
|---|---|---|---|---|---|---|
|  | N (%) | N (%) |  | N (%) | N (%) |  |
| Overall | **847 (84.19)** | **159 (15.8)** |  | **694 (69)** | **312 (31)** |  |
| • **Male** | 400 (84,7) | 72 (15,3) | 0,695 | 350 (74,2) | 122 (25,8) | <0.001* |
| • **Female** | 447 (83,7) | 87 (16,3) |  | 344 (64,4) | 190 (35,6) |  |
| Age |  |  |  |  |  |  |
| • **<20** | 20 (62,4) | 12 (37,5) | 0,003* | 14 (43,8) | 18 (56,3) | 0.028 |
| • **[20–30]** | 293 (81,2) | 68 (18,8) |  | 257 (71,2) | 104 (28,8) |  |
| • **[30–40]** | 326 (88,8) | 41 (11,2) |  | 250 (68,1) | 117 (31,9) |  |
| • **[40–50]** | 119 (85,6) | 20 (14,4) |  | 96 (69,1) | 43 (30,9) |  |
| • **≥50** | 89 (83,2) | 18 (16,8) |  | 77 (72) | 30 (28) |  |
| Profession |  |  |  |  |  |  |
| • **Health care worker** | 91 (91,9) | 8 (8,1) | <0,001* | 79 (79,8) | 20 (20,2) | 0.003* |
| • **Private sector worker** | 163 (89,6) | 19 (10,4) |  | 112(61,5) | 70 (38,5) |  |
| • **Public service personnel** | 128 (90,8) | 13 (9,2) |  | 109 (77,3) | 32 (22,7) |  |
| • **Retired** | 17 (100) | 0 (0) |  | 13(76,5) | 4 (23,5) |  |
| • **Student** | 207 (77,2) | 61 (22,8) |  | 183 (68,3) | 85 (31,7) |  |
| • **Teacher/Lecturer** | 108 (80) | 27 (20) |  | 97 (71,9) | 38 (28,1) |  |
| • **Others** | 133 (82,9) | 31 (17,1) |  | 114 (63) | 67 (37) |  |
| Working environment |  |  |  |  |  |  |
| • **At home** | 168 (82) | 37 (18) | 0,146 | 129 (62,9) | 76 (37,1) | 0.007* |
| • **Face to face interaction with customers** | 227 (83,8) | 44 (16,2) |  | 177 (65,3) | 94 (34,7) |  |
| • **Hospital** | 90 (88,2) | 12 (11,8) |  | 82 (80,4) | 20 (19.6) |  |
| • **Office** | 225 (87,5) | 32 (12,5) |  | 189 (73,5) | 68 (26.5) |  |
| • **Out door environment** | 137 (80,1) | 34 (19,9) |  | 117 (68.4) | 54 (31.6) |  |
| City of residence |  |  |  |  |  |  |
| • **Yaounde** | 538 (85,1) | 94 (14,9) | 0.009* | 439 (69.5) | 193 (30.5) | 0.023 |
| • **Douala** | 130 (89) | 16 (11) |  | 91 (62.3) | 55 (37.7) |  |
| • **Buea** | 70 (78,7) | 19 (21,3) |  | 56 (62.9) | 33 (37.1) |  |
| • **Others** | 109 (78,4) | 30 (21,6) |  | 108 (77.7) | 31 (22.3) |  |

*Statistically significant at $p < 0.05$

**Table 5. Associations between background characters and practice regarding COVID 19.**

| Variables | Practice Score (3–9) | Practice Score (0–2) | (P-value) |
|---|---|---|---|
| | N (%) | N (%) | |
| Overall | **612 (60.8)** | **394 (39.17)** | |
| **Gender** | | | |
| • Male | 307 (65) | 165 (35) | |
| • Female | 305 (57,1) | 229 (42,9) | (0,008)* |
| **Age** | | | |
| • <20 | 19 (59,4) | 13 (40,6) | |
| • [20–30] | 203 (56,2) | 158 (43,8) | |
| • [30–40] | 226 (61,6) | 141 (38,4) | (0,138) |
| • [40–50] | 91 (65,5) | 48 (34,5) | |
| • ≥50 | 73 (68,2) | 34 (31,8) | |
| **Profession** | | | |
| • Health care worker | 62 (62,6) | 37 (37,4) | |
| • Private sector worker | 109 (59,9) | 73 (40,1) | |
| • Public service personel | 86 (61) | 55 (39) | |
| • Retired | 16 (94,1) | 1 (5,9) | (0,030) |
| • Student | 159 (59,3) | 109 (40,7) | |
| • Teacher/Lecturer | 85 (63) | 50 (37) | |
| • Others | 95 (61,3) | 69 (38,7) | |
| **Working environment** | | | |
| • At home | 127 (62) | 78 (38) | |
| • Face to face interaction with customers | 144 (53,1) | 127 (46,9) | (0,004)* |
| • Hospital | 62 (60,8) | 40 (39,2) | |
| • Office | 164 (63,8) | 93 (36,2) | |
| • Out door environment | 115 (67,3) | 56 (32,7) | |
| **City of residence** | | | |
| • Yaoundé | 368 (58,2) | 264 (41,8) | |
| • Douala | 102 (69,9) | 44 (30,1) | (0,007)* |
| • Buea | 55 (61,8) | 34 (38,2) | |
| • Others | 87 (62,6) | 52 (37,4) | |

*Statistically significant at $p < 0.05$

## Discussion

COVID 19 is spreading rapidly across the whole world and increasing exponentially in Cameroon [4]. This is one of the first studies that identify symptoms (suspected cases) of COVID 19 which is a thoughtful thing to do in a population experiencing a sudden outbreak. This method used to recruit participants is cost-effective and feasible given that the data was collected during a period of confinement and it can be employed as a rapid screening method in subsequent pandemic situations. In this study, predominantly women in an overall 84.19% had adequate knowledge about the mode of transmission of COVID 19. Akwa et al. [9] reported in his study that > 80% of respondents knew the disease is transmitted by a handshake, person to person, and contact with infectious droplets only. Our findings show that there is an increase in the knowledge perception of disease transmission since respondents now know it is transmitted by touching contaminated surfaces and then touching eyes, nose, or mouth. We found that 69% of respondents had a high attitude score towards hospital seeking behavior while 60.8%

**Table 6.  Factors associated with knowledge and practice towards COVID 19.**

| Demographics/characteristics | Knowledge | P-value | Practice | P-value | Attitudes | P-value |
|---|---|---|---|---|---|---|
| | OR (95% CI) | | OR (95% CI) | | OR (95% CI) | |
| **Gender** | | | | | | |
| • **Male** | 1 | | 1 | | 1 | |
| • Female | 0,92 (0.66–1.30) | 0.65 | 0.72 (0.56–0.92) | 0.010* | 1.598(1.209–2.113) | 0.001* |
| Age | | | | | | |
| • <20 | 1 | | 1 | | 1 | |
| • [20–30] | 2.58 (1.21–5.54) | 0.015* | 0.88 (0.42–1.83) | 0.73 | 3.366(1.563–7.249) | 0.002* |
| • [30–40] | 4.77 (2.17–10.47) | <0,001* | 1.10 (0.53–2.29) | 0.81 | 2.746(1.190–6.339) | 0.018* |
| • [40–50] | 3.57 (1.51–8.42) | 0.004* | 1.30 (0.59–2.85) | 0.52 | 2.712(1.108–6.637) | 0.029* |
| • ≥50 | 2.97 (1.23–7.13) | 0.015* | 1.47 (0.65–3.32) | 0.36 | 3.347(1.310–8.553) | 0.012* |
| Profession | | | | | | |
| • Health care worker | 1 | | 1 | | 1 | |
| • Private sector worker | 0.75 (0.32–1.79) | 0.52 | 0.89 (0.54–1.47) | 0.65 | 0.418(0.234–0.748) | 0.003* |
| • Public service personnel | 0.87 (0.35–2017) | 0.76 | 0.93 (0.55–1.58) | 0.80 | 0.817 (0.430–1.551) | 0.536 |
| • Student | 0.30 (0.14–0.65) | 0.002 | 0.87 (0.54–1.40) | 0.57 | 0.614(0.166–2.273) | 0.465 |
| • Teacher/Lecturer | 0.35 (0.15–0.81) | 0.014 | 1.01 (0.59–1.73) | 0.96 | 0.622(0.331–1.166) | 0.138 |
| • Others | 0.42 (0.19–0.97) | 0.041 | 0.95 (0.57–1.57) | 0.83 | 0.385(0.212–0.698) | 0.002* |
| City of residence | | | | | | |
| • Yaounde | 1 | | 1 | | 1 | |
| • Douala | 1.42 (0.81–2.49) | 0.22 | 1.66 (1.13–2.45) | 0.010* | 0.715(0.480–1.064) | 0.098 |
| • Buea | 0.64 (0.37–1.12) | 0.12 | 1.16 (0.74–1.83) | 0.52 | 0.689(0.425–1.118) | 0.132 |
| • Others | 0.63 (0.40–1.01) | 0.053 | 1.20 (0.82–1.75) | 0.34 | 1.418(0.906–2.220) | 0.127 |

*Statistically significant at *p* <0.05

took the necessary precautions like avoiding crowded areas, wore masks, washing hands regularly as stipulated by the WHO and CDC guidelines [10]. The practice score was not as high as expected because 39.2% of people resorted to traditional concoctions and auto medications. These potentially risky behaviors were related to the female gender maybe because within Cameroon traditional context women are caregivers to the family. So communication initiatives, educational forums are needed to educate women on these risky practices.

The strict adherence to preventive practices could primarily be attributed to the very strict prevention and control measures implemented by local governments to put on masks and washing hands at every public place. The big question remains as to whether the masks are appropriately used given the current increase in the number of cases in Cameroon. The efficiency of social distancing may be compromised given that 26.9% of respondents interacted face to face (like in customer service points, banks, markets, etc), and the government later uplifted restrictive measures on bars and other leisure places.

The factors: gender, age, and city of residence which correlated positively with knowledge, practice for COVID 19 will be useful for public health policy-makers and health workers to recognize the target population for COVID-19 prevention and sensitization. The level of awareness on COVID-19 among Cameroonian residents was expected because 44.1% of the respondents knew that the disease outbreak was in December 2019 and 54.5% of the respondents got the information through the television before the first imported case in the country was recorded in March 2020. Sample characteristics such as; University students (BSc, Master, and Ph.D.) and private-sector workers, actively acquired knowledge of this infectious disease from various television channels, websites, and WhatsApp because of the alarming and global

**Table 7. Associations between age, gender with Comorbidity and symptoms of COVID 19.**

| Comorbidities | All patients N(%) | Gender | | P-value | Age (years) | | P-value |
|---|---|---|---|---|---|---|---|
| | | Female N(%) | Male N(%) | | <30 N(%) | ≥30 N(%) | |
| Hypertension | 47 (4.7) | 30 (5.6) | 17 (3.6) | 0.13 | 1 (0.3) | 46 (7.5) | <0.001* |
| Cancer | 2 (0.2) | 2 (0.4) | 0 (0.0) | 0.50 | 0 (0) | 2 (0.3) | 0.257 |
| Diabetes | 12 (1.2) | 8 (1.5) | 4 (0.8) | 0.39 | 1(0.3) | 11 (1.8) | 0.028 |
| Cardiovascular diseases | 7 (0.7) | 3 (0.6) | 4 (0.8) | 0.71 | 1 (0.3) | 6 (1.0) | 0.178 |
| Asthma | 19 (1.9) | 13 (2.4) | 6 (1.3) | 0.25 | 7 (1.8) | 12 (2.0) | 0.841 |
| Respiratory tract infection | 14 (1.4) | 8 (1.5) | 6 (1.3) | 0.75 | 3 (0.8) | 11 (1.8) | 0.27 |
| Common flu | 69 (6.9) | 37 (6.9) | 32 (6.8) | 0.93 | 26 (6.6) | 43 (7.0) | 0.81 |
| Allergic cough | 27 (2.7) | 19 (3.6) | 8 (1.7) | 0.068 | 7 (1.8) | 20 (3.3) | 0.16 |
| Tuberculosis | 4 (0.4) | 3 (0.6) | 1 (0.2) | 0.63 | 1 (0.3) | 3 (0.5) | 1.00 |
| Malaria | 247 (24.6) | 112 (21.0) | 135 (28.6) | 0.005 | 93 (23.7) | 154 (25.1) | 0.6 |
| **Symptoms of COVID19** | | | | | | | |
| Fever | 50 (5.0) | 26 (4.9) | 24 (5.1) | 0.87 | 21 (5.3) | 29 (4.7) | 0.063 |
| Dry cough/catarrah | 84 (8.3) | 38 (7.1) | 46 (9.7) | 0.13 | 27 (6.9) | 57 (9.3) | 0.174 |
| Headache | 132 (13.1) | 85 (15.9) | 47 (10.0) | 0.005 | 54 (13.7) | 78 (12.7) | 0.641 |
| Diarrhea | 9 (0.9) | 4 (0.7) | 5 (1.1) | 0.71 | 6 (1.5) | 3 (0.5) | 0.088 |
| Muscle pain | 60 (6.0) | 27 (5.1) | 33 (7.0) | 0.19 | 23 (5.9) | 37 (6.0) | 0.905 |
| Do not smell odor or taste | 22 (2.2) | 13 (2.4) | 9 (1.9) | 0.57 | 5 (1.3) | 17 (2.8) | 0.112 |
| Difficulty breathing | 25 (2.5) | 13 (2.4) | 12 (2.5) | 0.91 | 9 (2.3) | 16 (2.6) | 0.075 |
| Throat irritation | 65 (6.5) | 38 (7.1) | 27 (5.7) | 0.37 | 19 (4.8) | 46 (7.5) | 0.093 |

*Statistically significant at $p < 0.05$

situation of the epidemic. So good sources of communication among Cameroonian residents in pandemic situations like this are the television, social media followed by mouth to mouth communication by those who don't have access to these technologies. Cases of COVID 19 as with other diseases are broadly classified as suspected, probable, and confirmed cases [7]. Also, clinical symptoms vary from mild to moderate other than severe in old people with comorbidities [11]. Assessing the symptoms of COVID 19 (suspected cases) is a preliminary step in the diagnosis and management of this disease. This study showed that 41/1006 respondents had ≥ 3 symptoms (Fever, dry cough/catarrh, throat irritation, headache) linked to COVID 19. This estimate is hypothetical as some COVID 19 symptoms are equally clinical signs of other diseases like malaria and respiratory tract infections (bronchitis) [12]. Respondents experiencing ≥ 3 symptoms must seek medical help since the differential diagnosis of malaria or respiratory infections and COVID 19 is slim. Also, there are no significant differences in

**Table 8. Associations between symptomatology and KAP.**

| Symptomatology | Knowledge Score (4–7) N(%) | Knowledge Score (0–3) N(%) | P-value | Attitude Score (0–2) N(%) | Attitude Score (2–4) N(%) | P-value | Practice Score N(%) | Practice Score N(%) | P-value |
|---|---|---|---|---|---|---|---|---|---|
| **No Symptom** | 628 (84.8) | 113 (15.2) | 0.419 | 516 (69.6) | 225 (30.4) | 0.456 | 454 (61.3) | 287 (38.7) | 0.638 |
| **Symptoms** | 219 (82.6) | 46 (17.4) | | 178 (67.2) | 87 (32.8) | | 158 (59.6) | 107 (40.4) | |
| **0–3 symptoms** | 203 (83.2) | 41 (16.8) | 0.416 | 162 (66.4) | 82 (33.6) | 0.359 | 145 (59.4) | 99 (40.6) | 0.824 |
| **≥4 symptoms** | 16 (76.2) | 5 (23.8) | | 16 (76.2) | 5 (23.8) | | 13 (61.9) | 8 (38.1) | |

*Statistically significant at $p < 0.05$

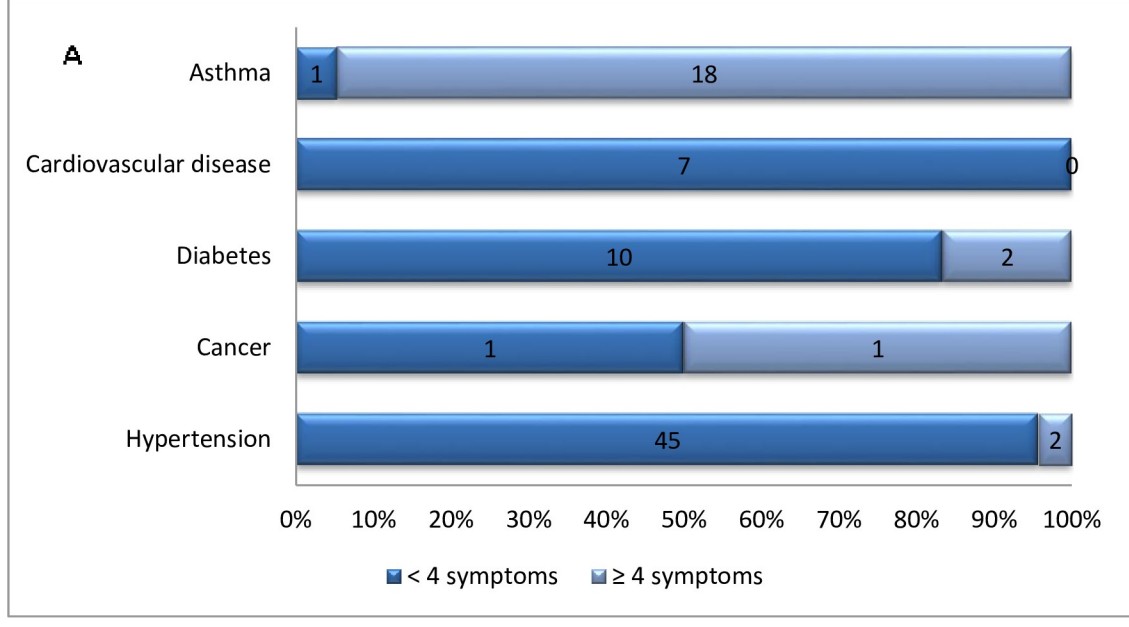

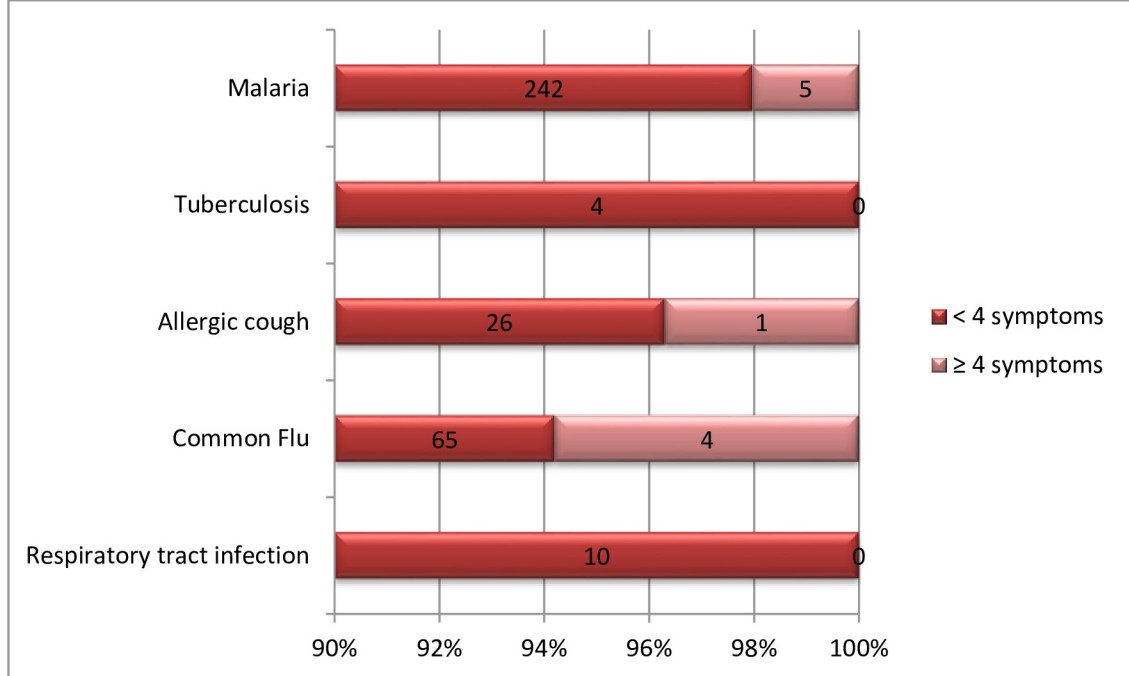

**Fig 1.** Associations between the number of respondents with A) comorbidities, B) diseases with some common symptoms to COVID 19 and <4 symptoms or ≥ 4 symptoms of COVID 19.

KAP between those having symptoms and those without symptoms. This warrants further investigation to assess the effectiveness of the measures put in place to curb the spread of the disease given the continuous increase in the number of cases and the overall high KAP score.

Also, hypertension was more prevalent in respondents > 30 years old with P-value <0.001. Arif et al [13] showed that hypertension is linked to old age with an overall prevalence of hypertension of 41.9% (95% CI: 37.2–46.6) in a total of 418 residents in Ethiopia ≥ 50 years

old. Overall 23 respondents having $\geq 4$ symptoms had underlying comorbidities and 10 of them had other diseases with symptoms similar to COVID 19 in our study. This could be potentially dangerous for these patients given that, previous studies on coronavirus death rates have also been shockingly higher in pre-existing comorbidity such as cardiovascular disease, Diabetes, Hypertension, Chronic respiratory disease, Cancer [14,15]. Of all the respondents,168 (18.1%) out of 262 of those with symptoms did not frequently or habitually experience such symptoms. Among the 168 respondents, those presenting with $\geq 3$ symptoms were considered as suspected cases.

The presence of $\geq 3$ symptoms in 4% (56% of them with co-morbidities) of the population surveyed supports the current trend in the number of confirmed cases (8681) in Cameroon. Thus widespread testing in the community is relevant because <22% of people with COVID 19 symptoms seek help. Given that 32(78.05%) of respondents who had more than 3 symptoms and did not call *1510*, won't be able to manage their conditions appropriately without the counsel of medical personnel they will further contribute to the spread of the disease. The hesitance to call was attributed to fear of getting contaminated, stigmatization if they are COVID 19 positive and misdiagnosis. Having symptoms and also comorbidities is high risk and not seeking help at the hospital and resorting to traditional concoctions with no standard dosage and auto-medications is much riskier. This calls for more sensitization and discrete ways of managing cases.

## Recommendations

COVID 19 outbreak has put the whole world under panic and in our context stigma. People's attitudes and practices could stem from panic and stigma or the knowledge provided to them. One way to avoid this could be to create a confidential online system to share COVID 19 experiences and consult online which is one of the objectives of our study. A better approach would have been to use more sophisticated software technics like qualtrics (which we didn't have in our settings) to geolocalize suspected cases and circumscribe a particular neighborhood for rapid and prompt intervention. Also, telecommunication industries could engage in sending daily consultation messages to the population on COVID 19 symptoms which could serve as baseline data for health personnel. The effectiveness of the prevention measures of COVID 19 still needs to be well established (total confinement being the best option) reason being that there is an increase in the number of cases regardless of a high KAP score observed in our study. Our study opens more doors for scholars who could use the same research design to collect data in similar situations; and learning from what is happening in Cameroon could be useful for comparative studies on COVID 19 experiences of other African countries. Women should be the primary target audience for behavior change initiatives by program managers on the management and understanding of COVID 19 disease. This behavior change initiative is of paramount importance to preclude negative attitudes of not going to the hospital (or calling 1510) when sick and encourage positive preventive and therapeutic practices, for fear of a rapid rise in mortality rate due to auto medications and traditional concoctions.

## Limitations of the study

Due to limited access to the internet and online information resources, populations in remote areas were not interviewed since the disease is most prevalent in the cities.KAP studies for people at the grass-root level in Cameroon are needed to assess their preparedness towards the COVID-19 pandemic. The second limitation is the limited sample representativeness used to assess suspected cases, comorbidities, and the unstandardized assessment of attitudes towards health-seeking behavior, which should be developed via focus group discussion and in-depth

interview accompanied by confirmatory tests. Thirdly, a KAP is a quantitative tool and that to focus on behavior change qualitative work would be necessary. Finally a poststratification analysis was not feasible in our study reason being that the latest census in Cameroon was carried out in 2005.

## Supporting information

**S1 Appendix. Survey questionnaire.**
(DOCX)

**S1 Table. City of residence of respondents.**
(DOCX)

## Acknowledgments

The authors express their profound gratitude to all Cameroonian respondents for their immense collaboration and all those who made the online transmission of the questionnaire possible.

## Author Contributions

**Conceptualization:** Adela Ngwewondo, Lucia Nkengazong, Marie Chantal Ngonde.

**Data curation:** Adela Ngwewondo, Jean Thierry Ebogo.

**Formal analysis:** Adela Ngwewondo, Lum Abienwi Ambe, Jean Thierry Ebogo, Fabrice Medou Mba, Hamadama Oumarou Goni.

**Investigation:** Adela Ngwewondo, Lucia Nkengazong, Lum Abienwi Ambe, Jean Thierry Ebogo, Fabrice Medou Mba, Hamadama Oumarou Goni, Nyemb Nyunaï, Marie Chantal Ngonde.

**Methodology:** Adela Ngwewondo, Lucia Nkengazong, Lum Abienwi Ambe, Jean Thierry Ebogo, Fabrice Medou Mba, Hamadama Oumarou Goni, Nyemb Nyunaï, Marie Chantal Ngonde.

**Project administration:** Adela Ngwewondo, Lucia Nkengazong, Nyemb Nyunaï, Marie Chantal Ngonde.

**Supervision:** Adela Ngwewondo, Lucia Nkengazong, Jean-Louis Essame Oyono.

**Validation:** Adela Ngwewondo, Lucia Nkengazong, Jean-Louis Essame Oyono.

**Visualization:** Adela Ngwewondo.

**Writing – original draft:** Adela Ngwewondo, Jean Thierry Ebogo.

**Writing – review & editing:** Adela Ngwewondo, Lucia Nkengazong.

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
