## [Decision Letter · Decision Letter 0]

12 Jul 2020

Dear Dr Adela,

Thank you very much for submitting your manuscript "Knowledge, Attitudes, Practices of/towards COVID 19 preventive measures and symptoms: A cross-sectional study during the exponential rise of the outbreak in Cameroon" for consideration at PLOS Neglected Tropical Diseases. As with all papers reviewed by the journal, your manuscript was reviewed by members of the editorial board and by several independent reviewers. In light of the reviews (below this email), we would like to invite the resubmission of a significantly-revised version that takes into account the reviewers' comments. 

We cannot make any decision about publication until we have seen the revised manuscript and your response to the reviewers' comments. Your revised manuscript is also likely to be sent to reviewers for further evaluation.

Sincerely,

Andrés Felipe Henao-Martínez, M.D.

Deputy Editor

Andrés Henao-Martínez

Deputy Editor

Reviewer's Responses to Questions

**Key Review Criteria Required for Acceptance?**

**Methods**

-Are the objectives of the study clearly articulated with a clear testable hypothesis stated?

-Is the study design appropriate to address the stated objectives?

-Is the population clearly described and appropriate for the hypothesis being tested?

-Is the sample size sufficient to ensure adequate power to address the hypothesis being tested?

-Were correct statistical analysis used to support conclusions?

-Are there concerns about ethical or regulatory requirements being met?

Reviewer #1: The objective as well as the hypothesis are both clearly stated. The challenge for the authors is to more clearly and strongly link the results to the hypothesis that the KAP can indeed impact the acceptance of recommended measures to interrupt transmission of the corona virus. One area that is not clear to me is the collection of information on symptomatology. This avenue of study did not clearly fit into the stated objectives as there is not effort made to tie individuals’ symptoms into the behaviors that a KAP seeks to establish. My recommendation is to look at whether self -reported symptoms plays a role in any of the three pillars of a KAP.

The study design is appropriate given that it was conducted during a time of partial confinement and self-quarantining. There are further comments on this in the discussion of the study’s limitations. The population is clearly described though (and again to be discussed later in the limitations section) the population is limited to urban dwellers possessing the necessary technology and are literate. This corresponds with the fact that apparently all respondents were employed with perhaps the exception of those listed under “other”. These factors are ones that may exclude those who are more vulnerable to disease i.e. those who are not as socially, economically advantaged. These factors should be noted in the study population and eligibility criteria section as well as in the discussion of limitations. 

I would also recommend that the authors provide more context of the disease within Cameroon citing available statistics and explicitly stating what measures the government has taken to slow transmission. This would be much more helpful than the global overview of the pandemic that is given. 

I defer the questions concerning sample size and statistical analysis to those who with more expertise than my own.

There are no ethical concerns though the authors could state if any recommendations and actions were made to those reporting symptoms that could indicate infection with the corona virus.

Reviewer #2: • The objectives stated are very superficial and no hypothesis are stated in the manuscript. It appears one of the attempts researchers made is to test the association of demographic characteristics on KAP. This part needs major revision of stating all the hypothesis testable from current data and articulating objectives.

• Study design to be improved stating dependent vs independent variables being studied with reference to each of the objective 

• It seems study sample is taken from 4 cities, of above 18 years age, with WhatsApp, email & website facility. Details of total population of the 4 cities viz. population size, salient demographic and social features, economic and educational status and incidence/prevalence of covid-19 etc be given. 

• Unless there is adequate justification to say this sample represents the general population of the 4 cities, the findings will be applicable to only to the sample category of the population. This may be one of the limitations.

 • For the current visible hypothesis sample size would be adequate. 

• The conclusions need to be rewritten based on the analysis which has to be in accordance with the objectives

To be mentioned briefly in the methodology: 

• Category of respondents, proportion of the sample (affected/healthy/suspected), Sampling procedure applied. 

• The activities mentioned in the manuscript like- suspecting the cases of covid-19, Assessing their symptoms, collecting the details of comorbidity etc are carried out as part this research study ? Whether primary or secondary methods used for every data collected. 

• Clarify possible range of min-max KAP scores. If the scores (mentioned in line 144) 3-7, 1-4, 2-9 are achieved ones it is possible there are some false questions, which may be deleted from manuscript as they will be misleading the readers.

Details of analysis done mentioning the types tables and statistical tests used for testing each of the two variables.

Reviewer #3: The paper employs an online survey with offline recruiting to study the Knowledge, Attitudes, and Practices related to the COVID-19 in Cameroon. The analysis is ok, and the interpretation of the results is proper. My only concern with methods are the following:

1. The recruiting is problematic: the respondents were recruited by a poster that invited people to answer the survey. This builds in a bias inside the responses, dismissed because the authors find the disease's prevalence similar to the reported number of cases. I believe this is still problematic, and the authors have to point out some problems and some strengths of this 'snowballing sampling' process.

Suggestion 1: Explain the recruiting process better and have an appendix with an example of the poster used for recruitment. Moreover, the word poster somehow gave me the idea that it was something physically placed somewhere. Luckily I read the paper twice, but some other people may have the same perception.

Suggestion 2: There certainly are strengths in snowballing a sample. The authors acknowledge that the sampling may be a limitation, but they could also discuss how it may be advantageous: cheap and feasible in a pandemics situation. This has to be stressed and perhaps suggested for similar studies.

2. The demographics: the authors should provide the demographics of the country and how they relate to their sample demographics. As their sample comes from people that know how to read, perhaps they should take the country demographics with these proper restrictions.

Suggestion: find out the country demographics and discuss how they relate with the survey one. One caveat here is that the last Census was taken in 2005. However, the authors could find projections in the BUCREP website (or at least explain why not possible to find reliable aggregated country-level data).

3. Answering time analysis: As the COVID grew over time, so did the media exposure and the knowledge from people. They had about one month of data collection going on. Did the patterns change from the first to the last 15 days? If so, how?

Suggestion: redo the analysis using the split sample of SPSS by dates, breaking in two or three subsequent periods.

4. Multilevel Regression and Post-stratification: Based on suggestions 2 and 3, you could also run a multilevel regression and poststratification on the covariates in a census to see if you could reweight the sample, to make the sample closer to the actual population in Cameroon.

Suggestion: check to see if you could apply multilevel regression, and have better estimates by each town or region. If not, at least raking the sample to make it closer to the proportions in the population. This piece might help to do the raking using SPSS (https://community.ibm.com/HigherLogic/System/DownloadDocumentFile.ashx?DocumentFileKey=17fd2f0b-7555-6ccd-c00c-5388b082161b&forceDialog=0).

Should these points be addressed, I believe the paper will increase considerably in quality and breadth.

Minor points:

1. What program did you use to collect the answers? If Qualtrics, you have georeferenced information from the respondent.

2. Have you incentivized the responses in any way?

3. I am not an expert on Cameroon, but is there any social media that could have been used to boost the response rates?

**Results**

-Does the analysis presented match the analysis plan?

-Are the results clearly and completely presented?

-Are the figures (Tables, Images) of sufficient quality for clarity?

Reviewer #1: The analysis plan is followed as far as it goes. The reader is presented with the findings of concerning knowledge, attitude and practices through tables that are sufficiently clear. I would recommend avoiding the terms of good and right and wrong knowledge and other somewhat judgmental adjectives. The fact that the global understanding of the virus is in flux, what is seen as “good” may not be in another month. The authors should present to us what their baseline is for judging behaviors and knowledge. An example is that whether hospital seeking behavior should be considered as a positive behavior considering that nosocomial transmission is indeed a risk.

As noted above, the symptomatology does not provide any additional insight into the results of the KAP. I would recommend that if possible, the authors analyze the symptomatology data in light of the other findings of the actual KAP. Does having symptoms impact on attitude, on practice, etc.?

Reviewer #2: Analysis needs major modifications, and also will have to be articulated as per the modified objectives and analysis plan:

Table-2 can bring out a crucial finding if it can be made a cross table (source vs period) to asses the magnitude of effective and early means of communications among the study population. 

Table-3: shows the response of total sample to the KAP questionnaire. No need to show the question as such in the table. Need to provide N value wherever only % is given. 

• Table-4 seems to be primarily used for testing the association between level of knowledge of the total sample and demographic variables, which needs percentage corrections. Cross tabulations with independent variables in rows need to show row percentages, so that as in case of gender (table-4) it can be inferred, the proportion of respondents with high or low knowledge is more or less from which gender. Column percentages presently showing only the gender proportion, not of those with high and low knowledge in each gender. In the same row the test used for significance and (p= ) value details etc be mentioned. Mean scores has no importance. 

• Similar tables be developed for testing any associations if in the objectives between any variables and the levels attitudes, practices, Symptomatology, Comorbidity with title specifying the association being tested in the table. 

• Avoid using of value judgments by using ‘low knowledge(0-3)’ and ‘high knowledge(4-7)’ or adequate knowledge ( ) and inadequate knowledge( ) mentioning the score ranges instead of ‘right knowledge’ and ‘wrong knowledge’ in table-4. Similarly avoid using terms like ‘good practice’ & ‘wrong practice’ in table-5; positive and negative practices or risky or non risky /traditional and modern practices which are used in discussion may be used.

• All the tables like table-7, should show totals or (N= ) in proper place . From page-15 under symptomology need to specify the symptoms of what disease. 

Findings from each of the figures and tables be clarified as per objectives.

Reviewer #3: The results are clear and intelligible. The tables and images are of excellent quality. The interpretations are precise and match the results in the tables. The writing is clear and concise, and I thank the authors for having written such a pleasant paper to read.

**Conclusions**

-Are the conclusions supported by the data presented?

-Are the limitations of analysis clearly described?

-Do the authors discuss how these data can be helpful to advance our understanding of the topic under study?

-Is public health relevance addressed?

Reviewer #1: I find the discussion and conclusion sections to be in need of greater thought particularly in regard to the public health implications. I congratulate the authors for undertaking this work rooted in behavioral health as these are the elements that prior to the development of therapeutic drugs and vaccines that will reduce the infection and mortality rate. The data provided offers a number of avenues to explore within a public health context which need much more critical analysis. For example, In the last line of the first paragraph in the discussion section, it was stated that …”potentially risky behaviors were related to the female gender’’’”. This finding (which also needs to be better justified) opens up a number of doors for interventions with women as the target audiences whether it is communication initiatives, educational approaches, etc. The authors suggest a confidential on-line system to share experiences. In making this recommendation, the data from the KAP should be used by the authors to justify the idea. 

My overall recommendation is for the authors to tie the data more directly into interventions that could be recommended to governmental and non-governmental organizations.

In terms of limitations, I think the authors overlook a number of issues here. Primarily, the use of the technology limits participation to those who have access to the various platforms used. Another limitation is the study depends on respondents being literate which also excludes many of Cameroon’s poor and as we see in various countries disparities along socio-economic lines of who gets infected are very clear. The relatively high % of people taking recommended precautionary measures could easily be related to their educational level, income, etc. 

Though the disease may initially be more prevalent in cities, we have seen in other settings the migration to home villages bringing infection with them. Their recognition of the need for further work at the grass roots level is well taken and I would recommend it being a recommendation in the conclusion or discussion section.

Additionally, I would suggest that if the authors are not prepared to recommend interventions based on the results, they state that a limitation to the study is that a KAP is a quantitative tool and that to really focus on behavior change qualitative work would be necessary.

The final limitation of the study is the constant flux in the global knowledge of Covid-19 and what is considered best practices.

Reviewer #2: The conclusion- People’s attitudes and practices could stem from panic and stigma- to be verified whether it is out of stigma or provided knowledge to them. 

Rest of the conclusions are suggestions to the programme, addressing them with online education receiving facility and who visit hospitals. Both seem to be out come of the responses. 

The authors' discussions about KAP and raise in cases can not be made as 2nd time data not available. 

The knowledge, attitude and Practices of the study sample can not be attributed the general population of Cameroon, Findings of the study sample , represent the behaviour a better community with on line communication facility. 

However, as per the findings there are 41/1006 respondents had ≥ 3 symptoms; Provided out of them 32(78.05%) of respondents who had more than 3 symptoms and did not call 1510, this research may help to estimate how many more cases with symptoms and not positive to report in a complex of several communities of Cameroon, where there are confirmed cases 8681, 4836 recovered and 212 deaths.

Reviewer #3: The authors made a compelling case for why we should consider their data as important information to have wide-spread testing in Cameroon. They also claimed that the data suggest a few improvements that could be made, especially in the gender-related correction of misperceptions about the COVID. I agree with them that this is relevant to public health.

However, it would have been interesting to discuss a bit about how we can apply the knowledge acquired in Cameroon for other African countries.

Suggestion: improve the conclusion with recommendations, first, of how scholars could use the same research design to collect data in similar situations; and second, how learning what is happening in Cameroon helps us to understand the COVID experiences of comparable countries.

**Editorial and Data Presentation Modifications?**

Reviewer #1: The paper needs copy-editing to ensure the authors’ intentions are being clearly communicated. Some comments follow:

Methodology 

Line 22 – not clear “interacted face-to-face

Line 24 – positive needs to be defined as well as good

Conclusion:

Line 28 (1st sentence of paragraph) needs clarification

Line 29 – Information that would be welcome in Background

Summary:

Lines 46 – use of satisfactory and good 

Line 49 – needs to be clarified

Introduction

Line 61 – delete “seems to be very contagious”

Line 66 – the phrase “ Some of these measures…” is insufficient. If there are others beyond what is listed, the reader would want to know them. If this is the complete list, delete “Some of”

Line 72 – The sentence needs some wordsmithing – ex The infection rate and ths the resources needed to battle this disease can be expected to increase exponentially

Lines 74-83 – I would consider deleting most of this discussion of treatments as much of it is in flux. The main point here is that without treatments available many may turn to “non-standard options”

Study Design

Line 110 – what other cities/what populations

Line 116 – restricted to literate respondents

Symptomatology

There needs to a clearer link between the KAP and reported symptoms. Otherwise it doesn’t really belong in this paper. Why are we interested in the symptoms. Was there any effort to explore their knowledge in terms of what symptoms are associated with CVD-19.

Discussion

Last line of the first paragraph: this seems very speculative while at the same time does not really explain why women may report riskier behaviors. If the authors want to pursue this line of thinking, a more in-depth look is warranted. If correct and more clearly justified, it does provide program managers with a primary target audience to for behavior change initiatives

The last paragraph explores more deeply the issues of signs and symptoms and implications than KAP but if possible should be analyzed in light of the KAP objectives stated in the beginning.

Reviewer #2: (No Response)

Reviewer #3: (No Response)

**Summary and General Comments**

Reviewer #1: As noted elsewhere, the authors have taken a bold step to look at the current pandemic from a behavioral perspective understanding the import that knowledge, attitudes and practices will play in reducing the transmission rate and lowering mortality. Prior to publishing, I think the paper requires further thinking as to how the findings can advance Cameroon’s efforts to limit the impact of the infection. I believe the data is there but more than just presenting the data is needed. A more robust discussion of interventions is needed as defined by the results of the study. If symptomatology is to be included, it needs to be linked to the stated objective of the paper, i.e. assess the knowledge, attitudes and practices. Anchoring the paper more firmly in the Cameroon context, worrying less about the global situation, will also be helpful to fully understand how to move forward.

Reviewer #2: • The title may be slightly modified to convey the main features of the study-whether it is on health population or affected peoples.

The concepts used in the study viz. Knowledge/perception, Attitudes, Practices, Symptomatology and co-morbidities may briefly be defined and explained from the scope of the current study. 

Table-1: show (N=1006)

• In the profession- who all include others. housewives are part of the study or not.

• In working environment-instead of Face to face interaction with customers their place of work may be mentioned. 

• Face to face interaction (Yes/No) may be given below if data is available.

• Following variables may be added: Education and family income/economic status, Disease related data (healthy/with early signs/confirmed of Covid-19)

Reviewer #3: In this paper, the authors study the Knowledge, Attitudes, and Practices regarding the COVID-19 using an online survey in Cameroon. The authors recruited, from April 20 to May 20, 1006 respondents, using social media and websites, and the recruitment page directed the respondents to an online survey comprised of 32 questions. They find that respondents have a consistent knowledge about the COVID pandemics and that their symptom check and comorbidity studies match the prevalence of COVID-19 in the Cameroonian population. They also find that women tend to be less informed about COVID, which is interesting, as, around the world, women seem to care more about the disease than men.

Although the paper has several merits, I suggest a few revisions that would improve the article if adequately carried out. In the methods section, I detail my suggestions, guiding what I think could be done to improve the piece.

PLOS authors have the option to publish the peer review history of their article (what does this mean?). If published, this will include your full peer review and any attached files.

Reviewer #1: Yes: Chad MacArthur

Reviewer #2: Yes: MOTURU SOLOMON RAJU

Reviewer #3: No
---

## [Editor Report · Decision Letter 1]

11 Aug 2020

Dear Dr Adela,

We are pleased to inform you that your manuscript 'Knowledge, Attitudes, Practices of/towards COVID 19 preventive measures and symptoms: A cross-sectional study during the exponential rise of the outbreak in Cameroon' has been provisionally accepted for publication in PLOS Neglected Tropical Diseases.

Best regards,

Andrés Felipe Henao-Martínez, M.D.

Deputy Editor

Andrés Henao-Martínez

Deputy Editor

---

## [Editor Report · Acceptance letter]

28 Aug 2020

Dear Dr Adela,

We are delighted to inform you that your manuscript, "Knowledge, Attitudes, Practices of/towards COVID 19 preventive measures and symptoms: A cross-sectional study during the exponential rise of the outbreak in Cameroon," has been formally accepted for publication in PLOS Neglected Tropical Diseases.

Best regards,

Shaden Kamhawi

co-Editor-in-Chief

Paul Brindley

co-Editor-in-Chief
